# Mixed hierarchical local structure in a disordered metal–organic framework

Adam F. Sapnik [1], Irene Bechis[2], Sean M. Collins [1,3], Duncan N. Johnstone [1], Giorgio Divitini [1], Andrew J. Smith [4], Philip A. Chater [4], Matthew A. Addicoat [5], Timothy Johnson [6], David A. Keen [7], Kim E. Jelfs [2] & Thomas D. Bennett [1✉]

Amorphous metal–organic frameworks (MOFs) are an emerging class of materials. However, their structural characterisation represents a significant challenge. Fe-BTC, and the commercial equivalent Basolite® F300, are MOFs with incredibly diverse catalytic ability, yet their disordered structures remain poorly understood. Here, we use advanced electron microscopy to identify a nanocomposite structure of Fe-BTC where nanocrystalline domains are embedded within an amorphous matrix, whilst synchrotron total scattering measurements reveal the extent of local atomic order within Fe-BTC. We use a polymerisation-based algorithm to generate an atomistic structure for Fe-BTC, the first example of this methodology applied to the amorphous MOF field outside the well-studied zeolitic imidazolate framework family. This demonstrates the applicability of this computational approach towards the modelling of other amorphous MOF systems with potential generality towards all MOF chemistries and connectivities. We find that the structures of Fe-BTC and Basolite® F300 can be represented by models containing a mixture of short- and medium-range order with a greater proportion of medium-range order in Basolite® F300 than in Fe-BTC. We conclude by discussing how our approach may allow for high-throughput computational discovery of functional, amorphous MOFs.

[1] Department of Materials Science and Metallurgy, University of Cambridge, Cambridge, UK. [2] Department of Chemistry, Imperial College London, Molecular Sciences Research Hub, White City Campus, London, UK. [3] School of Chemical and Process Engineering & School of Chemistry, University of Leeds, Leeds, UK. [4] Diamond Light Source Ltd, Diamond House, Harwell Campus, Didcot, Oxfordshire, UK. [5] School of Science and Technology, Nottingham Trent University, Clifton Lane, Nottingham, UK. [6] Johnson Matthey Technology Centre, Blount's Court, Sonning Common, Reading, UK. [7] ISIS Neutron and Muon Facility, Rutherford Appleton Laboratory, Harwell Campus, Didcot, Oxfordshire, UK. ✉email: tdb35@cam.ac.uk

Metal–organic frameworks (MOFs) are hybrid materials composed of metal centres bridged by organic linkers. Our understanding of their structure–property relationships enables the rational design of functional materials through the selection of structural building units with specific chemistries and connectivities. This design-based approach has led to the rapid expansion of the MOF field, offering emerging solutions for challenges faced in gas storage and separation, catalysis and drug delivery[1,2]. Amorphous MOFs retain the local structural building units of their crystalline counterparts but do not possess the long-range order associated with crystallinity[3]. They have displayed great promise in the trapping of harmful guest species, improved ion transport capacity and tuneable drug delivery[3–5]. Furthermore, recent investigations into the liquid and glassy states of MOFs have revealed the potential for permanent porosity in these materials[6–9]. Amorphous MOFs are typically obtained via structural collapse or melt-quenching of a crystalline parent material, with very limited reports of direct syntheses[3].

The structures of amorphous MOFs are notoriously challenging to characterise. The absence of structural periodicity precludes the use of single crystal diffraction techniques routinely employed in the structure solution of crystalline materials. Since it is the atomic-scale structure of materials, often regardless of the degree of long-range order, that drives functionality, the comparatively poor understanding of amorphous MOFs prevents their rational design and consequently hinders their development towards possible applications.

Local structure probes are often the only means for characterising the structure of amorphous MOFs[3]. Spectroscopic techniques such as infrared, nuclear magnetic resonance and X-ray absorption spectroscopy are able to characterise linker bonding modes, chemical environments and coordination geometries. However, these techniques are limited by their spatial resolution, which is often restricted to nearest-neighbour atomic distances. Total scattering techniques probe the short- and long-range structure of a material through the simultaneous interpretation of Bragg and diffuse scattering obtained from a diffraction experiment[10]. The Fourier transform of the total scattering data yields the pair distribution function (PDF)—a real-space mapping of the two-body atom–atom correlations. The use of PDF methods to study disordered materials has become increasingly prominent in recent years due to its sensitivity towards deviations from the average structure[11,12].

While models of amorphous inorganics, such as amorphous silicon and germanium, date back over 50 years, the structural modelling of amorphous MOFs remains in its infancy by comparison[13]. The majority of studies have focused on the zeolitic imidazolate framework (ZIF) family, with amorphous ZIF-4 ($a$ZIF-4, $Zn(C_3H_3N_2)_2$) being the prototypical example. In fact, of the 205 models recently curated into a database of porous, rigid, amorphous materials, $a$ZIF-4 was the only MOF-based entry to be included[14]. This entry was from a study of $a$ZIF-4 using neutron and X-ray total scattering data with reverse Monte Carlo (RMC) modelling to demonstrate a reconstructive transition upon heating from the ZIF-4 crystal and the topological similarity of $a$ZIF-4 to the continuous random network structure of silica[15]. Subsequent efforts to produce atomistic models of $a$ZIF-4 involved simulating the melt-quenching process using classical molecular dynamics or using ab initio molecular dynamics to provide mechanistic insight[8,16]. Neither of these latter two techniques are readily generalisable towards other MOF architectures; the former requiring reactive force fields whilst the latter remains very computationally taxing and is currently limited to small configuration sizes.

More recently, an alternative approach adapted from the simulation of amorphous organic polymers was used to overcome these challenges[17]. Employing a generalised polymerisation algorithm, a structure for $a$ZIF-4 was produced in the absence of experimental total scattering data[18]. The structural building blocks of ZIF-4 were polymerised together to generate an amorphous model through a process that was entirely independent of the crystal symmetry of crystalline ZIF-4. Steps within the algorithm generated chemical bonds between reactive sites of the metal nodes and organic linkers, not too dissimilar to the automated assembly of the secondary building unit method used in the de novo prediction of crystalline inorganic structures over 20 years ago[19]. This eliminates the requirement for a generalised MOF reactive force field, which has been a major barrier towards modelling the structures of amorphous MOFs in recent years. The comparatively minimal initial parametrisation, experimental input and computational power required for this approach makes it a particularly appealing route to generate structures of amorphous MOFs, especially given the often-challenging measurements required to characterise these materials. Despite the advantages of polymerisation-based modelling, to the best of our knowledge it has currently only been used to produce a model for the $a$ZIF-4 structure, though this was not experimentally verified using pair distribution function data nor compared to the RMC-derived model for $a$ZIF-4. The true predictive nature and generality of this approach towards other amorphous MOFs has yet to be explored.

Our focus here is Fe-BTC, an Fe (III) based MOF containing 1,3,5-benzenetricarboxylate linkers. A form of Fe-BTC is one of the very few commercially available MOFs, known as Basolite® F300. It has garnered interest due to its broad catalytic ability, revered as possessing 'the most general and high catalytic activity' of the four commercial MOFs in 2012[20]. For example, as a Lewis acid catalyst in Claisen-Schmidt reactions, ring opening of epoxides and selective hydrogenations, and as a catalyst in the oxidation of thiols and amines, achieving conversion rates and selectivities of 99% in some cases[20]. Despite this, the atomic-scale structure of Fe-BTC remains unknown. The material is typically obtained via a sol-gel route, with subsequent drying or annealing leading to the formation of a morphologically diverse family of materials, all known as Fe-BTC[21]. For example, drying via exchange with supercritical carbon dioxide affords FeBTC aerogels that retain hierarchically porous micro- and macrostructure[21]. Conversely, harsher drying conditions lead to the formation of a denser, less porous, xerogel or even powdered Fe-BTC samples. While in other reports, dry-gel conversion of Fe-BTC induces a phase transition to a crystalline framework[22]. Consequently, the exact atomic microstructure of Fe-BTC is particularly poorly understood, previously being described as disordered, amorphous and nanocrystalline[23–26]. It is surprising, given the intrinsic links between structure and catalytic ability, that the atomic-scale structure of Fe-BTC is still unknown.

All Fe-BTC materials share a broad, ill-defined, powder diffraction pattern that prevents determination of the precise atomic positions. Local structure investigations using extended X-ray absorption fine structure (EXAFS) suggested that Fe-BTC shares structural similarity to MIL-100, a crystalline framework with the same chemical composition[27]. MIL-100 possesses a hierarchical structure based on $FeO_6$ octahedra that cluster around a shared oxo-anion to form a trimer unit (Fig. 1a)[28]. Four of these trimers assemble via the organic linker to form a tetrahedron (Fig. 1b). These form the nodes upon which the **mtn** topology network is constructed (Fig. 1c). The cubic unit cell (space group $Fd\bar{3}m$) of MIL-100 is complex, containing over 10,000 atoms, with a unit cell volume in excess of 390,000 Å$^3$ (Fig. 1d). The EXAFS study revealed Fe-BTC shares the trimer motif with MIL-100[27]. Structural investigations into Fe-BTC have typically been hindered by the short-range length scales associated with local structure probes. This limited understanding of the local structure has prevented attempts to elucidate an atomistic model for Fe-BTC.

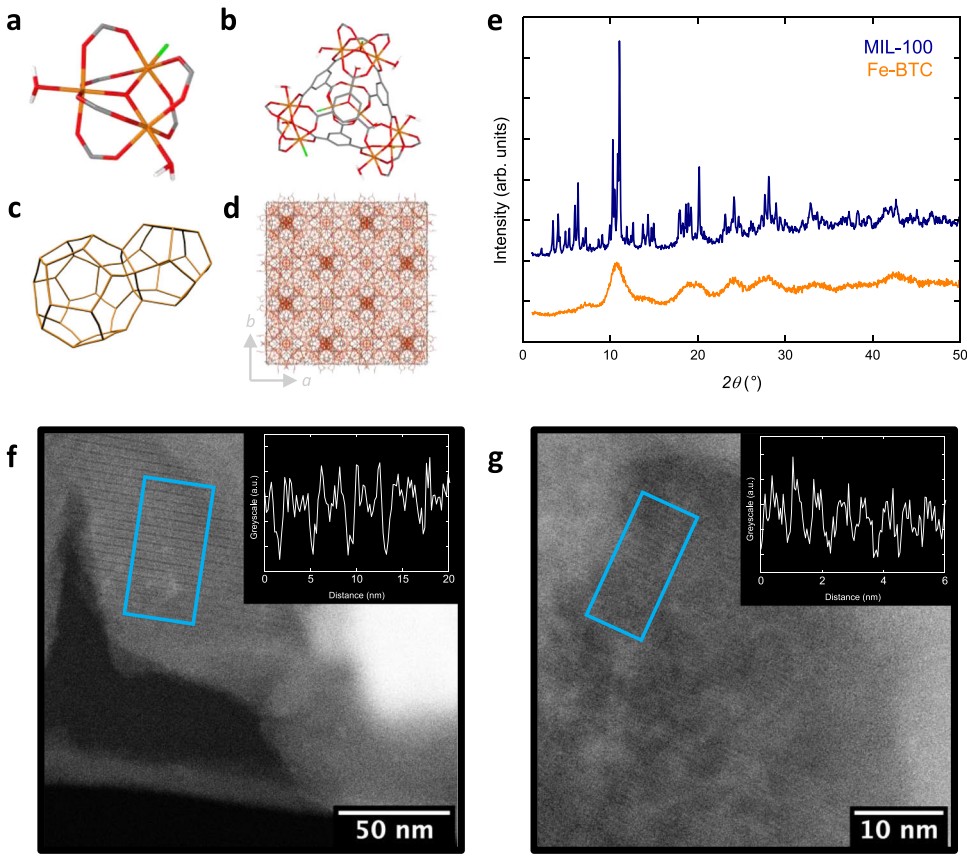

**Fig. 1 Structural characterisation of MIL-100 and Fe-BTC. a** Oxo-centred trimer unit (Fe orange, O red, C grey, F green and H from ligated water molecules white). **b** A tetrahedron assembled from four oxo-centred trimer units via the organic linker. **c** Dual pore structure of the **mtn** framework, tetrahedral units form the nodes of the connected network. Each vertex represents the centroid of a tetrahedron. **d** Crystallographic unit cell of MIL-100. **e** Powder X-ray diffraction data for MIL-100 (blue) and Fe-BTC (orange), data offset for clarity. **f** High-resolution scanning transmission electron micrograph of MIL-100. Lattice fringes corresponding to structural periodicity are highlighted in the blue box. The inset shows the corresponding line profile of the lattice fringes highlighted. The distance between minima is 4.2 nm and corresponds to the (111) crystallographic planes of the MIL-100 structure. These features correspond to the Bragg reflections at 2.1° in (**e**). Data reproduced from ref. [34]. **g** High-resolution scanning transmission electron micrograph of Fe-BTC. Lattice fringes are highlighted in the blue box. The inset shows the corresponding line profile of the lattice fringes highlighted. The distance between minima is 0.71 nm, consistent with a shorter degree of ordering compared to MIL-100. These features correspond to the scattering centred around 12.5° in (**e**).

In this study, we address these challenges using a combined experimental and computational approach. We use a combination of electron, X-ray and neutron-based characterisation techniques to probe the structure of Fe-BTC. We employ advanced high-resolution scanning transmission electron microscopy to identify a nanocomposite structure of Fe-BTC containing crystalline nanoparticles embedded within an amorphous matrix. Synchrotron total scattering measurements highlight the similarity in local structure between Fe-BTC and MIL-100 and reveal the presence of tetrahedral assemblies within Fe-BTC. Applying our understanding of the local structure, we build an atomic-scale model of Fe-BTC using an adapted polymerisation algorithm. Our model is the first demonstration of this computational approach outside of the well-studied ZIF family. We anticipate this computational approach will be key in enabling the discovery of the many thousands of amorphous MOF structures that are likely to exist and will subsequently help guide our future experimental efforts.

## Results

**Structural characterisation.** Powder X-ray diffraction measurements from MIL-100 displayed sharp Bragg scattering, consistent

with its expected crystalline nature (Fig. 1e). Pawley refinement of the diffraction data confirmed no additional phases were present (Supplementary Fig. 1 and Supplementary Table 1). In comparison, only broad, weak, scattering was observed for Fe-BTC, which is characteristic of an amorphous or nanocrystalline material[3,29]. Notably, the peaks in the scattering were centred around the regions of strong Bragg scattering present in MIL-100. The exception to this was below 10°, which corresponds to the largest *d* spacings in MIL-100, where featureless scattering was observed for Fe-BTC. The identical chemistry, alongside the qualitative similarity in the positions of scattering, alludes to some degree of structural similarity between the two materials and points toward a potentially amorphous or nanocrystalline structure for Fe-BTC.

For nanocrystalline materials, the inverse relationship between crystallite size and diffraction peak width is understood through the Scherrer equation[30]. For small domain sizes, Bragg peaks may appear very broad and can easily be mistaken for an amorphous material[29]. Although Pawley refinement of the Fe-BTC diffraction data was possible using the MIL-100 crystallographic unit cell following the inclusion of a crystallite size term, at these small domain sizes it is challenging to draw reliable conclusions from such analysis and additional structural probes are required to

delineate between a potentially nanocrystalline or amorphous material (Supplementary Fig. 2 and Supplementary Table 2).

Scanning electron microscopy was used to probe the morphologies of MIL-100 and Fe-BTC and, in particular, to look for nanoparticles in Fe-BTC. Micrographs of MIL-100 revealed 20 μm aggregates of sub 1 μm particles, consistent with the synthetic route used (Supplementary Fig. 3)[25]. While some degree of faceting was apparent, it was not possible to determine the exact morphology of the microcrystalline particles. Fe-BTC on the other hand appeared inhomogeneous and contained two morphologically distinct phases; large fragments on the order of 20 μm and small, irregular particles around 200 nm (Supplementary Fig. 4). The surfaces of the larger domains were studded with the smaller particles, akin to MOF composite materials where crystalline MOF particles are embedded into an amorphous or glassy matrix[31,32]. Visual inspection of the images indicated the majority of the sample comprised of the larger fragments, suggesting this is the dominant phase.

The (111) crystallographic planes of MIL-100 have a particularly large $d$ spacing of 4.2 nm, a characteristic signature of the large unit cell[28]. MIL-100 and Fe-BTC samples were probed using combined synchrotron small- and wide-angle X-ray scattering (SAXS and WAXS) at the I22 beamline (Diamond Light Source, UK) to investigate whether the nanoparticles present in Fe-BTC were in fact domains of MIL-100. The SAXS data were relatively featureless, which is as expected, as any small-angle scattering will be augmented by intra-particle length scales present in the powdered samples [Supplementary Fig. 5]. The exception to this was the 4.2 nm $d$ spacing that was clearly seen in MIL-100. No corresponding feature was observed in Fe-BTC, consistent with the lack of Bragg scattering in the lab-source data at low $2\theta$. This absence suggests the nanoparticles in Fe-BTC are not domains of MIL-100. This was further corroborated in the WAXS data, where a significant proportion of other low-$Q$ Bragg peaks in MIL-100 were not present in Fe-BTC (Supplementary Fig. 6). The majority of peaks present in Fe-BTC were broadened compared to MIL-100, there were two exceptions to this (peaks at 0.3 and 1.5 Å$^{-1}$) that remained fairly sharp, which is consistent with an anisotropic structure of the crystallites. The qualitative similarity in scattering at higher $Q$ is however indicative of structural likeness on a shorter length scale. We note that this does not rule out the possibility of crystalline nanoparticles entirely, it simply establishes the absence of domains of nanocrystalline MIL-100. As an additional validation of this, the Pawley refinement of MIL-100 given in Supplementary Fig. 1 was convoluted with an isotropic, Gaussian broadening term constrained to 200 nm (i.e. the approximate size of the nanoparticles observed in the SEM data) (Supplementary Fig. 7). The calculated diffraction pattern contained much sharper Bragg scattering than observed in the experimental Fe-BTC data. Hence, if the nanoparticles in Fe-BTC were single crystal domains of MIL-100, then clear Bragg scattering would be expected.

With this nanocomposite structure in mind, high-resolution scanning transmission electron microscopy (HR-STEM) data were collected at ePSIC (Diamond Light Source, UK) to probe the structural differences between the phases in Fe-BTC and detect the presence of crystallinity. When operated at suitably low doses (electron fluence), HR-STEM enables the observation of structural periodicities in MOFs in the form of lattice fringes[33]. These give an indication of the length scale associated with periodicity in a material. Micrographs of MIL-100 exhibited clear examples of lattice fringes that indexed to the characteristic 4.2 nm $d$ spacing (Fig. 1f and Supplementary Fig. 8). These were observable for one data frame; thereafter the sample exhibited signs of electron beam-induced changes to the structure (i.e. beam damage).

Observations of lattice fringes were less prevalent in Fe-BTC, either due to the inherent lower proportion of crystalline material or potentially increased beam sensitivity of the sample. These periodic features were typically observed over a much smaller field of view than in MIL-100, within the region of 10–50 nm wide, and likely correspond to the nanoparticles observed in the scanning electron microscopy. No lattice fringes corresponding to a $d$ spacing of 4.2 nm were observed in the images collected, consistent with its absence in the SAXS data. Instead, a range of smaller $d$ spacings were observed, the largest of which was only 0.71 nm—signifying a smaller unit cell of the crystalline component in Fe-BTC compared to MIL-100 (Fig. 1g and Supplementary Fig. 9). Regions of crystallinity were often neighboured by domains lacking visible lattice fringes which, whilst not definitive, is suggestive of an absence of periodicity: that is, an amorphous phase corresponding to the bulk material supporting the nanoparticles. At the interfacial region between phases, the contrast was often mottled, and clear boundaries were hard to identify suggesting intimate phase interactions. Collectively, the electron microscopy data suggest a nanocomposite structure of Fe-BTC containing two distinct phases: an amorphous matrix representing the vast majority of the material, and crystalline nanoparticles of a phase other than MIL-100.

Fourier-transform infrared (FT-IR) spectroscopy data were collected to compare the linker bonding modes in Fe-BTC and MIL-100. Spectra acquired from Fe-BTC and MIL-100 contained two bands between 1445 and 1370 cm$^{-1}$ and two between 1630 and 1564 cm$^{-1}$, corresponding to the symmetric and asymmetric stretches of the carboxylate group, respectively (Supplementary Fig. 10). These four bands are characteristic of the linker bonding in a *syn-syn* fashion[28]. The carboxylate asymmetric stretching region in Fe-BTC was broader than in MIL-100 and may be indicative of an additional contribution from a monodentate asymmetric carboxylate stretch, which typically occurs at 1550 cm$^{-1}$ [34,35]. This suggests a degree of unsaturation of the linkers in Fe-BTC. In other words, it is probable that metal–linker bonding defects are present in the structure. We cannot, however, conclusively attribute the monodentate carboxylate signal specifically to the nanocrystalline (likely at the particle surface) or the amorphous phase.

Thermogravimetric analysis was used to explore the effect of the metal–linker defects in Fe-BTC on its thermal stability compared to MIL-100. Both materials showed mass losses associated with the removal of guest molecules and structural water prior to decomposition (Supplementary Fig. 11). Firstly, losses of 9.69 and 5.97 wt.% at *ca*. 100 °C for MIL-100 and Fe-BTC, respectively, are ascribed to the removal of guest water molecules. While the hydration from atmospheric water was not controlled, this may be indicative of a less porous interior within Fe-BTC. Secondly, water molecules coordinated directly to metal centres were lost at around 200 °C. MIL-100 exhibited a 1.85 wt.% loss whilst Fe-BTC showed a slightly higher loss of 2.10 wt.%. This minor difference suggests a higher degree of structural water in Fe-BTC. This is consistent with the presence of metal–linker defects resulting in additional coordinatively unsaturated metal sites. Finally, a sharp loss of 51.09 wt.% corresponding to thermal decomposition is observed at 365 °C for MIL-100. Decomposition of Fe-BTC begins at 310 °C and terminates at 460 °C, via a two-step process. Firstly 26.31 wt.% loss was observed, followed by a 34.40 wt.% loss—a 60.71 wt.% loss in total. The decomposition of Fe-BTC may result from either some unreacted linker or, as we have previously reported, metal–linker defects[24,34]. The two-step decomposition route may also arise from different thermal stabilities of the nanocrystalline and amorphous phases. After decomposition, bright orange hematite (Fe$_2$O$_3$) residue was obtained[34]. The iron content in MIL-100 and Fe-BTC was

estimated as 26.1 and 21.8 wt.%, respectively, assuming hematite to be the only solid product remaining after heating to 850 °C, which is close to the expected 25.7 wt.% calculated from the empirical formula for MIL-100.

Nitrogen adsorption isotherms were collected at 77 K to compare the porous nature of Fe-BTC and MIL-100 (Supplementary Fig. 12). As suggested by the TGA analysis, we found Fe-BTC to be less porous than the giant-pore MIL-100 structure. MIL-100 displayed a total pore volume of 0.991 cm$^3$ g$^{-1}$ and a BET surface area of 2240 m$^2$ g$^{-1}$, consistent with previous reports (Supplementary Table 3)[25]. Fe-BTC was essentially non-porous to nitrogen displaying a total pore volume of 0.035 cm$^3$ g$^{-1}$ and BET surface area of 6 m$^2$ g$^{-1}$. As previously discussed, different methods of drying Fe-BTC result in the formation of morphologically different phases (e.g. aerogels, xerogels and powders)[21]. The particular synthetic approach employed here utilises a high reaction concentration and more aggressive drying conditions, resulting in a low porosity material. For comparison, Basolite® F300 was reported by the manufacturers to possess a BET surface area between 1300 and 1600 m$^2$ g$^{-1}$, though independent experimental reports observed a surface area of around 840 m$^2$ g$^{-1}$ [36]. Porosity is heavily influenced by atomic structure, hence the wide range of porosities is a further indication of the structural diversity within the Fe-BTC family of materials[37]. Helium pycnometry measurements revealed similar densities for both MIL-100 and Fe-BTC, 2.03 and 1.91 g cm$^{-3}$, respectively. Elemental (CHN) analysis revealed that MIL-100 and Fe-BTC have very similar chemical compositions, containing 31.2 and 33.6 wt.% of carbon, respectively which is in line with the theoretical 33.1 wt.% calculated from the empirical formula for MIL-100 (Supplementary Table 4).

**Pair distribution function analysis**. Total scattering measurements and PDF analysis were carried out to compare the atomic configurations in the local structure of MIL-100 and Fe-BTC. Synchrotron X-ray total scattering data were collected using the I15-1 beamline (Diamond Light Source, UK). The structure factor for MIL-100 displayed many Bragg scattering features consistent with its crystalline nature (Fig. 2a). The intense feature at 0.7 Å$^{-1}$ corresponds to a cluster of Bragg peaks centred around the (733) crystallographic plane and represents a $d$ spacing of ~8.96 Å. Fe-BTC displayed a noticeable decrease in intensity of the peaks compared to MIL-100, with the Bragg peak cluster at 0.7 Å$^{-1}$ significantly less intense (Fig. 2a inset). While the structure factor for Fe-BTC neither clearly reflects a highly crystalline, nor an amorphous system, the data do however point towards an appreciable degree of order. Therefore, Fe-BTC exhibits a degree of order between an extended crystal and an entirely amorphous solid. This is consistent with the nanocomposite structure—containing both nanocrystalline and amorphous components—revealed by the electron microscopy data.

Following the necessary corrections, the Fourier transform of the total scattering data was calculated to obtain the real-space PDF (Fig. 2b). The crystalline nature of MIL-100 was immediately apparent from the long-range (>20 Å) correlations in the PDF. The partial PDFs calculated from the crystal structure of MIL-100 were used to assign the major contributions to the first three intense peaks at 1.98, 3.34 and 4.68 Å (labelled i to iii): (i) Fe–O, (ii) Fe–O and Fe–Fe and (iii) Fe–O, Fe–C and C–O (Fig. 2b and Supplementary Fig. 13)[23,38]. Beyond this, peaks within the PDF arise from multiple overlapping contributions. In order to interpret the PDF beyond the first few peaks, we compared the experimental data to calculated PDFs from the isolated building blocks of MIL-100's hierarchical structure (i.e. the trimer unit and tetrahedral assembly). Calculation of the PDF from a trimer unit

and a tetrahedral assembly revealed that correlations in the MIL-100 PDF between 1 and 7 Å largely originate from the local structure of the trimer, while correlations between 7 and 12 Å are predominantly from the tetrahedra (Fig. 2c, d)[34]. Hence, we deduce that correlations beyond 12 Å arise from the periodic ordering of tetrahedral units.

In Fe-BTC, the PDF contains contributions from both the amorphous matrix and the crystalline nanoparticles, and indeed from interactions between the two phases. While the amorphous matrix may not contain any long-range order, the bonding between neighbouring atoms will follow fundamental chemical principles and hence the local arrangement of the atoms will not be entirely random[23,39]. Consequently, the local structure of the amorphous matrix will contribute to the PDF at low-$r$ despite its disordered nature. The similarity between Fe-BTC and MIL-100 is immediately apparent when comparing the two PDFs (Fig. 2b inset). In particular, the PDFs are essentially identical up to 7 Å, thus confirming the presence of the trimer units within Fe-BTC. Interestingly, there is substantial similarity within the region 7–12 Å as well. This indicates that assembly of the trimers into tetrahedra is occurring to some degree within Fe-BTC. Previous studies on Fe-BTC using EXAFS and laboratory PDF measurements were limited by their real-space range and were unable to detect the presence of tetrahedra[23,27]. Here, the use of synchrotron radiation for total scattering measurements enables the local structure to be investigated over a greater length scale. Further comparison revealed no additional correlations are present in Fe-BTC compared to MIL-100; the local structure of Fe-BTC can be described in terms of MIL-100-like building blocks. Hence, the amorphous matrix and nanocrystalline phases of Fe-BTC possess similar local structure.

The relative intensity of peaks in the tetrahedral (7–12 Å) to trimer (1 and 7 Å) regions is lower in Fe-BTC relative to MIL-100. Using the most intense peaks at *ca.* 2 Å and 9.6 Å as simple representatives for the trimer and tetrahedral regions, respectively, noting the similarity in peak widths, a simple peak analysis was carried out. The ratio of the trimer:tetrahedra peak intensity in MIL-100 is 1:0.424, while in Fe-BTC it is only 1:0.167. From the crystal structure of MIL-100 it is clear that all trimer units are present in the form of the tetrahedral assemblies[28]. In other words, there are no free trimers in the structure. Assuming the ratio of 1:0.424 in MIL-100 corresponds to a 100% assembled structure, this suggests that only 39% of trimers are present in Fe-BTC in the form of tetrahedra. Repeating this analysis with the second most intense peaks in the trimer and tetrahedral regions at *c.a.* 3.3 and 8.0 Å, respectively, reveals the proportion of trimers in the form of tetrahedra in Fe-BTC is likely within the range of 39–64%. Therefore, Fe-BTC contains a considerable proportion, between 36 and 61%, of free trimers that are not arranged into tetrahedral assemblies. Of course, given the presence of both crystalline and amorphous domains in Fe-BTC, these values only represent an average, and it is likely that there will be a range of microscopic heterogeneity within the material. The observed damping of Fe-BTC's PDF at higher $r$ may also arise due to the partial nanocrystallinity of the material. Nanocrystallinity results in the increase of diffraction peak width in reciprocal space, which manifests as a damping of the peak heights in the PDF as a function of increasing $r$ in real space[10]. An alternative interpretation may include the formation of partial tetrahedra, those which do not contain all four constituent trimer vertices, within Fe-BTC. However, the geometric frustration of such defects is likely to come with a significant energetic penalty and so we consider the presence of fully formed tetrahedral assemblies as the most plausible explanation.

Beyond 12 Å there were very few correlations in Fe-BTC; those present are over-emphasised by the $D(r)$ formalism of the PDF[40].

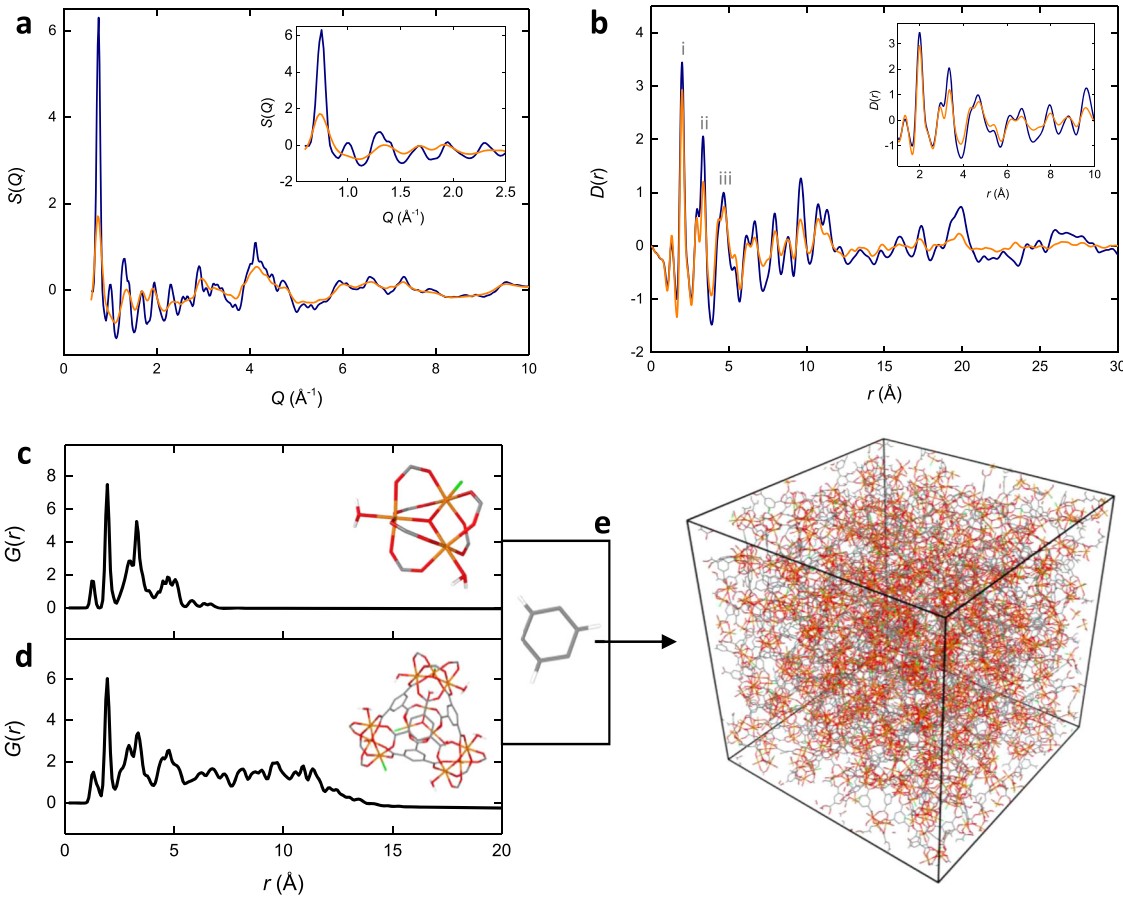

**Fig. 2 Pair distribution function analysis. a** X-ray structure factor for MIL-100 (blue) and Fe-BTC (orange), inset shows the low-$Q$ region. Total scattering data on MIL-100 reproduced from ref. [34]. **b** X-ray PDF for MIL-100 (blue) and Fe-BTC (orange), obtained via the Fourier transform of the data in (**a**). Inset shows low-$r$ region. **c** Calculated PDF for a single trimer unit, structure shown inset. **d** Calculated PDF for a single tetrahedral assembly, structure shown inset. **e** An amorphous model is obtained following the polymerisation of the metal nodes and linker, saturation and annealing steps. Hydrogens have been omitted.

Therefore, the PDF suggests a structure in which there is little coherence beyond the dimensions of the tetrahedral assemblies. This is consistent with the bulk amorphous matrix composition of Fe-BTC, where correlations are intrinsically absent at high-$r$ due to disorder. Through analysis of the PDFs, we have established the local structure of Fe-BTC contains both isolated trimer units and trimers assembled into tetrahedra in an approximately 1:1 trimer to tetrahedra ratio. Following our characterisation of the local structure, we sought to capture the bulk amorphous, atomic structure of Fe-BTC using a polymerisation-based computational approach.

**Structural modelling**. Polymatic, a polymerisation-based algorithm, was used to generate atomistic models of the amorphous phase that represents the bulk of Fe-BTC. Polymatic allows for the polymerisation of monomer units—metal nodes (trimer units and tetrahedral assemblies) and 1,3,5-benzenetricarboxylate anions in this case—into amorphous models[17]. Briefly, a simulation box was randomly populated with monomers and reactive sites within a defined cut-off distance were allowed to bond. After each bond formation, the structure was relaxed, and the process repeated until no further reactive sites could be found within the established cut-off distance. Finally, unreacted sites were capped, and the structure was annealed using an established 21-step molecular dynamics protocol (Fig. 2e). A total of approximately 91.9% of reactions were completed in these simulations in

comparison to 98.6% obtained for $a$ZIF-4, this reduction was due to the increased complexity and size of the structural building units in Fe-BTC compared to ZIF-4[18]. The force field employed here was verified to reproduce well the structure of crystalline MIL-100 and its building blocks[41,42]. Guided by the PDF analysis, we used this approach to produce three models of Fe-BTC with varying degrees of local structure. Specifically, a short-range order model (SRO) containing 100% trimers, a mixed model (MIX) containing a 50/50 mixture of trimers and tetrahedra, and a medium-range order model (MRO) containing 100% tetrahedra. Five independent models were built for each of the three systems, in order to provide good statistical sampling of the material. For additional details on the model generation and validation, see Supplementary Methods.

The porosity of each system (SRO, MIX and MRO) was analysed using a nitrogen-sized probe (1.82 Å radius) within the Zeo++ software package (Fig. 3a and Supplementary Table 5)[43]. The results were averaged across the five simulations and the standard deviations given in parentheses. Both the SRO and MIX models contained only non-accessible to nitrogen surface areas of 243(36) and 392(36) $m^2 g^{-1}$, respectively, which would not be detected using a nitrogen adsorption experiment. The MRO model possessed both accessible (724(77) $m^2 g^{-1}$) and non-accessible (78(28) $m^2 g^{-1}$) surface area, demonstrating the increase in surface area with the proportion of tetrahedral assemblies in disordered Fe-BTC materials. Visualisation of these nitrogen-probed surface areas revealed pore structures that are

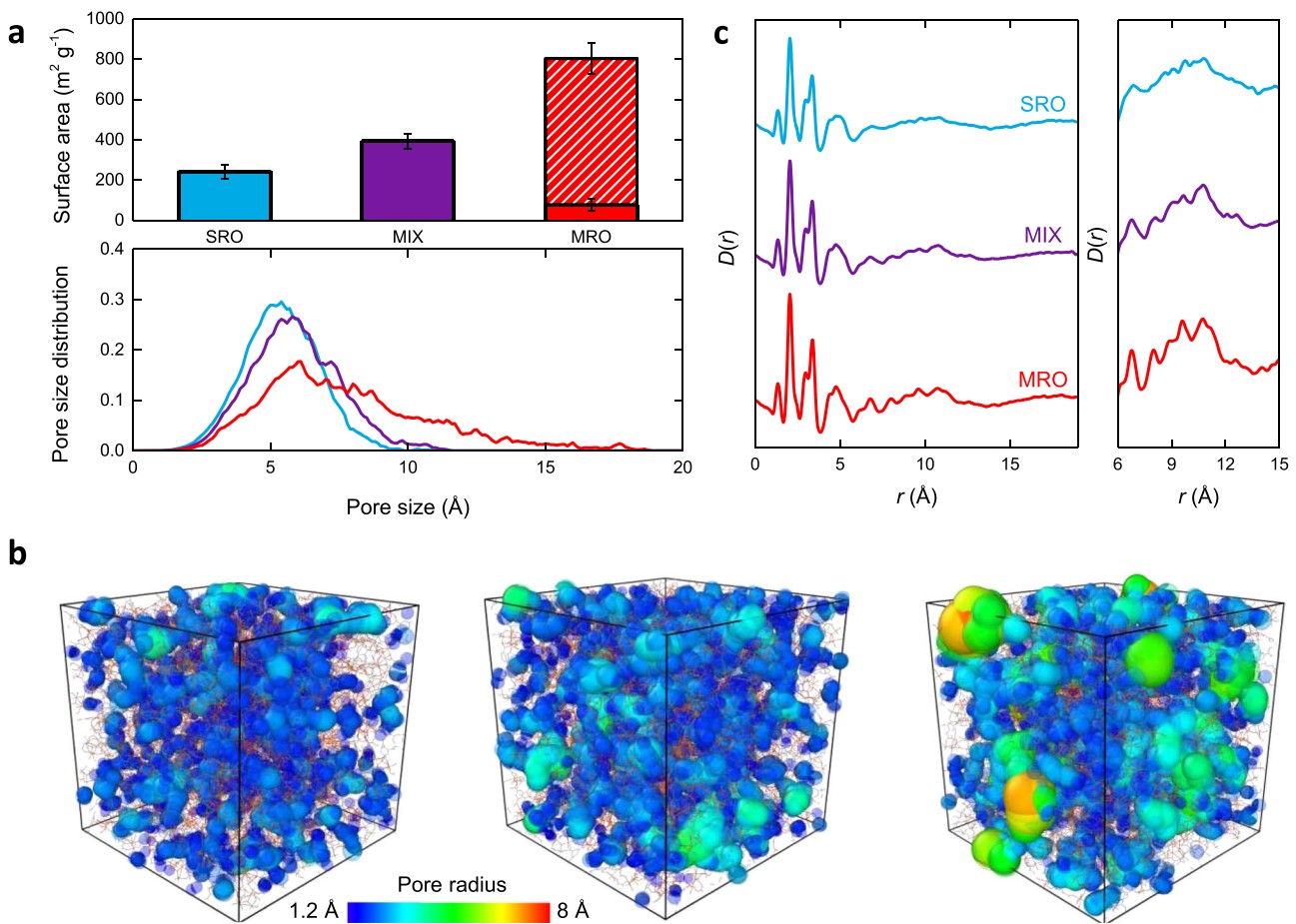

**Fig. 3 Amorphous model analysis. a** Top: accessible (striped) and non-accessible (filled) surface areas to a nitrogen probe (radius 1.82 Å) for all three amorphous model types; only the MRO model contains accessible surface area. Error bars show the standard deviation calculated from five packings of each model type. Bottom: average pore size distribution for the SRO (blue), MIX (purple) and MRO (red) models, averaged over the five simulations. **b** Visual representation of the pore size distribution for the SRO (left), MIX (centre) and MRO (right) systems from a representative model of ~70 Å in length. **c** Calculated PDFs for the SRO (blue), MIX (purple) and MRO (red) models. Right-hand panel shows the region corresponding to the tetrahedra, with the intensity scaled by a factor of three. Correlations in this region become more apparent as the proportion of tetrahedra in the model is increased.

twisted and irregular in shape (Supplementary Fig. 14). All three models exhibited broad distributions of pore sizes, typical of an amorphous material—similar pore size distributions were observed for aZIF-4 (Fig. 3a)[18]. The maximum size of the pores in the structure increases as the proportion of tetrahedra is increased (i.e. SRO < MIX < MRO) (Fig. 3b). The critical window size is defined as the maximum probe size that can percolate through the unit cell from one side to the other. As anticipated, the average critical window size in the models also increases with the proportion of tetrahedra, 2.94(0.11) [SRO], 3.19(0.10) [MIX] and 4.92(0.66) Å [MRO]. This explains the absence of nitrogen-accessible surface area in both SRO and MIX, with the kinetic diameter of nitrogen being 3.64 Å[44]. From a purely geometric perspective, these critical window sizes indicate the capacity for ion transport, such as lithium (1.8 Å diameter) or sodium (2.32 Å diameter) ions, may be possible in these models[18].

The calculated PDFs for the SRO, MIX and MRO models appear identical up to 7 Å due to the common trimer building block present in all models (Fig. 3c and Supplementary Fig. 15). Subtle, yet crucial, differences are present in region 7–12 Å. The SRO model is very broad and largely featureless in this region, consistent with the model only containing trimer units and hence no medium-range order. The MIX and MRO models, which contain tetrahedral building blocks, both possess peaks in this

region, with these being more intense in the MRO model (Fig. 3c right-hand panel).

Peak analysis of the experimental Fe-BTC PDF suggested that Fe-BTC contained both individual trimer units and tetrahedral assemblies, in an approximately 1:1 ratio. Based on this analysis, we anticipated the MIX model to best represent the average structure of Fe-BTC, though such a small model is unlikely to capture the full structural variety present in Fe-BTC. Comparison between the SRO model and Fe-BTC confirmed this model does not sufficiently capture the local structure of Fe-BTC; several peaks are observed in the region 7–12 Å of Fe-BTC that are not present in the calculated data (Supplementary Fig. 16).

Both the MIX and MRO models capture the atomic structure in the 7–12 Å region of the PDF. However, the MRO model is significantly more porous than experimentally observed and it is only the MIX model that also reproduces the observed lack of porosity in Fe-BTC. The incorporation of trimer units in the MIX model allows for increased structural disorder that prevents accessibility to a nitrogen probe. Hence, we deduce that—based on PDF and porosity analysis—the MIX model best reflects the atomic-scale structure of Fe-BTC's amorphous matrix as prepared here (Fig. 4a, b). The physical realisation of the MIX structure can be rationalised in terms of the rapid reaction kinetics. Formation of trimer units precedes their assembly to

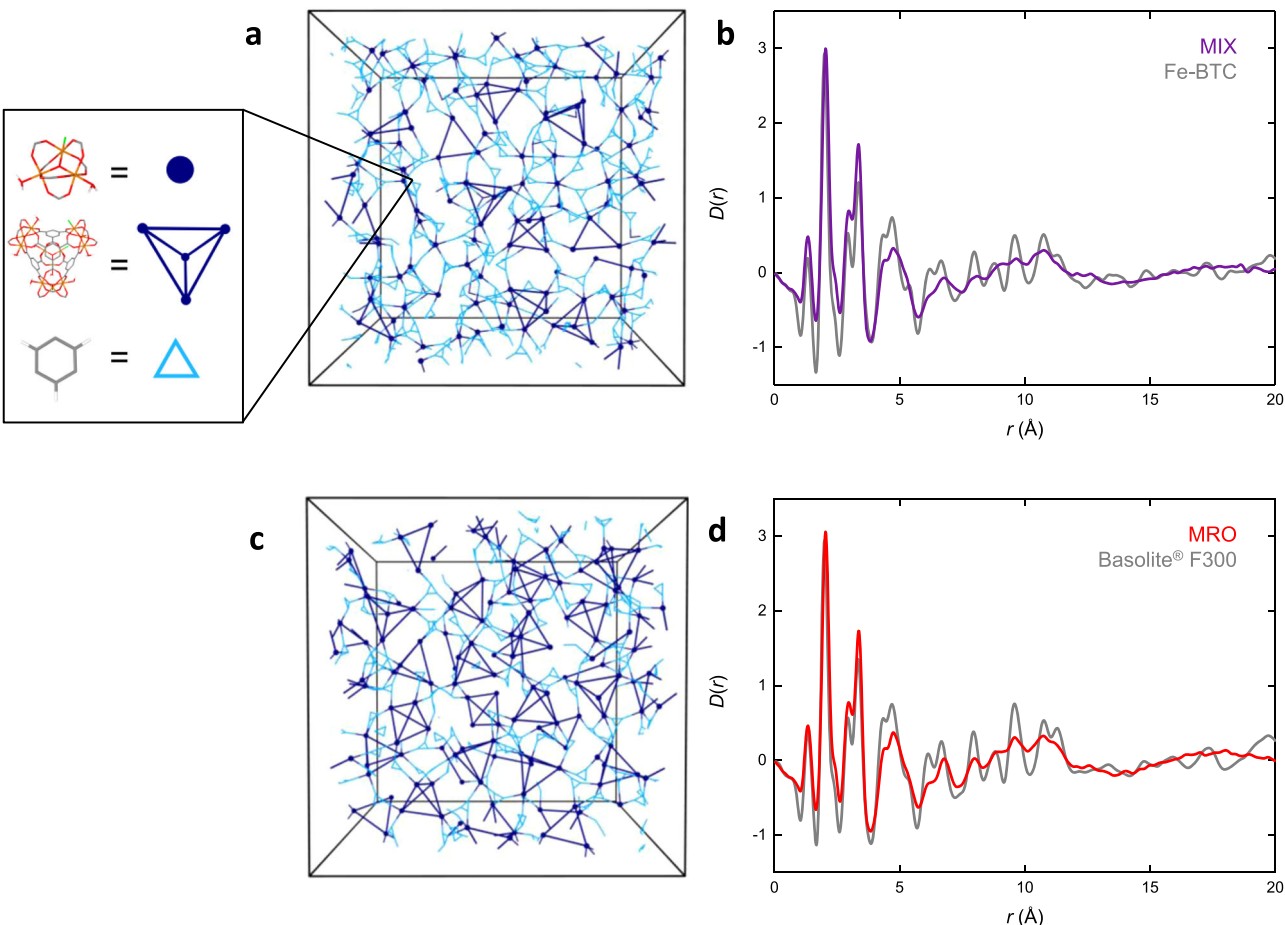

**Fig. 4 Structure of Fe-BTC. a** Schematic diagram of the MIX model. A representative 20 Å slice of the model is shown for clarity. Dark blue circles represent the centre of the trimer units. Trimer units within a tetrahedral assembly are connected via dark blue lines. Organic linkers connecting trimer units and tetrahedral assemblies (excluding those within the tetrahedra themselves) are shown as light blue triangles. Inset shows the relationship between the model and schematic. **b** Experimental X-ray PDF for Fe-BTC (grey) and calculated PDF for the MIX model (purple). **c** Schematic model of the MRO model. In the MRO model, all trimers are assembled into tetrahedra. **d** Experimental X-ray PDF for Basolite® F300 (grey) and calculated PDF for the MRO model (red).

form tetrahedra.Given the gelation of Fe-BTC occurs within seconds of mixing, it is likely that crosslinking of trimer units *via* the organic linker in a disordered fashion competes with tetrahedral assembly resulting in a combination of the two in the final structure.

The models generated using our approach result entirely from the simulation and are not refined using experimental data as would be the case in RMC approaches, hence the agreement between the simulation and experiment is very promising. As an additional validation of the selected MIX model, a deuterated sample of Fe-BTC was synthesised and neutron total scattering data were collected (ISIS Neutron and Muon Source, UK). Powder X-ray diffraction measurements of the deuterated sample revealed similar scattering compared to Fe-BTC, except for an additional, low angle region of scattering around 4° which may suggest a slightly greater degree of order in the deuterated sample [Supplementary Fig. 17]. The neutron PDF for Fe-BTC showed reasonable agreement with the calculated PDF from the MIX model, peak positions were largely similar between the experimental and calculated data [Supplementary Fig. 18]. The small rise in intensity between 3.1 and 10.4 Å in the experimental data is believed to result from the additional low angle scattering in the deuterated sample.

Our understanding of the structure–property relationship between the proportion of tetrahedra and porosity is important

for our understanding of another key member of the Fe-BTC family, namely Basolite® F300. The structure of Basolite® F300 remains unclear, however, it exhibits a broad powder X-ray diffraction pattern almost identical to our Fe-BTC sample, with some suggestion of additional low angle scattering (Supplementary Fig. 19). Basolite ®F300 is significantly more porous than Fe-BTC as synthesised here, with a reported surface area of 840 m$^2$ g$^{-1}$ [36]. We, therefore, proposed the MRO model (with accessible surface area of 724(77) m$^2$ g$^{-1}$) as a potential candidate for the structure of Basolite® F300. In order to validate this, X-ray total scattering data on Basolite® F300 were collected (Supplementary Fig. 20). Comparison between the experimental PDFs of Fe-BTC and Basolite® F300 showed they were almost identical (Supplementary Fig. 21). In particular, we find that Basolite® F300 contains the peaks corresponding to the presence of trimer and tetrahedral structural building blocks. The crucial difference lies in the relative intensity of peaks in the 7–12 Å region. Performing the same peak analysis used previously on Fe-BTC reveals a greater proportion (between 60 and 90%) of tetrahedral assemblies in Basolite ®F300 compared to Fe-BTC (between 39 and 64%), consistent with its increased porosity. Therefore, from the model types (SRO, MIX and MRO), it is the MRO model that best reflects the structure of Basolite® F300 in terms of PDF and porosity analysis. Comparison between the calculated and experimental PDFs of the MRO model and Basolite® F300 shows a similar level of agreement to Fe-BTC

and the MIX model (Fig. 4c, d). Crucially, this demonstrates the versatility of the computational approach employed in this study. By varying parameters within the simulation, we are able to model —with no experimental input—the structures of different Fe-BTC materials to obtain good agreement with experimental PDF and porosity data.

## Discussion

The use of advanced electron microscopy identified that Fe-BTC possesses a nanocomposite structure, containing both crystalline and amorphous domains. At the same time, synchrotron PDF analysis revealed the presence of tetrahedral assemblies in Fe-BTC. This understanding enabled the use of a polymerisation-based algorithm to generate several atomistic models of Fe-BTC, containing different degrees of tetrahedral units. This, in turn, allowed a key structure–property relationship between the degree of tetrahedral assembly and porosity to be deduced. A model containing both independent trimer units and tetrahedral assemblies reproduced the experimental Fe-BTC porosity and PDF data, while that containing only trimers within tetrahedral assemblies is proposed as a model for the structure of the more porous, industrially significant Basolite® F300.

The polymerisation-based model is the first demonstration of this methodology to the amorphous MOF field outside of the prototypical aZIF-4 structure, and provides a clear foundation for future fundamental and applied studies on Fe-BTC[18]. On a structural level, our immediate goal would be to characterise the crystalline domains to obtain a model of the atomic structure of these regions through electron diffraction measurements for example. Subsequent combination of the crystalline model with that of our amorphous phase presented here would provide a model for the full nanocomposite structure of Fe-BTC. This model could then serve as a starting configuration for a large-box refinement against the experimental data. Both stages represent significant challenges; structure solution within composite materials is very difficult and has yet to be reported using electron diffraction measurements, which are typically limited to single crystal samples[45]. Furthermore, while two-phase modelling of nanocomposite structures is possible, the RMC methods required are complex[39]. In particular, structural relaxation at the interfacial boundary between phases and energetic relaxation of the final configuration are both very challenging, yet exciting, avenues for future exploration. We note that while the models here represent the bulk amorphous structure, it is possible that there may also be phase mixing amongst the three model types, even on a microscopic scale. There is, of course, an entire continuum of configurations between the extremes of our SRO and MRO models. On a more applied footing, the geometric analysis undertaken here on Fe-BTC suggests its suitability for ion transport, while the structure will be useful in understanding and improving the catalytic ability of Fe-BTC by helping to identify the presence of active sites for example.

Crucially, the ability to model amorphous MOFs has significance well beyond the example of Fe-BTC. This study emphasises the necessity for multifaceted approaches towards characterising potentially disordered or amorphous systems, and particularly where, like here, both may coexist. In particular, local crystallinity probes such as HR-STEM are highly complementary to X-ray diffraction and PDF measurements[33,46]. From a computational perspective, polymerisation-based modelling to generate amorphous MOF structures has been demonstrated. The comparatively minimal parametrisation, experimental input and computational power required by this approach may be advantageous over other techniques for modelling amorphous materials. The approach may be used to generate amorphous models

with different structural and physical properties, through variation in metal node configurations, or by artificially reducing the extent of polymerisation during the simulation, in order to incorporate additional defects into the structure. This is especially appealing given the rise of defect engineering strategies that can readily augment the properties of functional hybrid materials[47].

A general approach towards modelling amorphous MOFs will prove pivotal in developing the future of the field. Recently, Yaghi et al. addressed the significance of a digital revolution in the reticular design-based approach that has driven the development of the crystalline MOF field[48]. It is only a matter of time before such tools are applied to the rapidly expanding amorphous hybrid material field. For example, we can envisage a database of amorphous MOF models obtained through this polymerisation-based methodology. Libraries of metal node configurations and organic linkers can be created and combined in an essentially infinite number of configurations to obtain amorphous MOF structures. The resulting database can then be screened for desirable properties, a concept analogous to those curated for crystalline MOF architectures, such as the CoRE MOF database or the recent more general database of porous rigid amorphous materials[14,49–53]. These insights can then be used to drive subsequent experimental efforts towards the realisation of new amorphous hybrid materials. Importantly, characterising these amorphous materials ultimately aids our understanding of crystalline MOFs too, given that amorphous intermediates are observed for many prototypical crystalline MOF systems such as ZIF-8, UiO-66 and MOF-5[54–57].

We anticipate that much of the functional diversity afforded by crystalline MOF structures will translate over to their amorphous analogues, for example, photocatalysis in titanium-containing frameworks[58], luminescent behaviour in lanthanide materials[59], and antimicrobial properties in silver containing structures[60]. Understanding the atomic-scale behaviour of amorphous MOFs will enable the elucidation of their underlying structure–property relationships and reinforce the idea that whilst structure drives functionality in MOFs, it is not necessary for that structure to be crystalline.

## Methods

**Synthesis**. All chemicals were used as received with no further purification. Iron (III) nitrate nonahydrate (99.95%), 1,3,5-benzenetricarboxylic acid (95%), methanol (99.8%), methanol-OD (99.8% D), deuterium oxide (99.9% D), ethanol (99.8%), ammonium fluoride (99.99%), sodium hydroxide pellets (98%), iron (II) chloride tetrahydrate (99.99%) and Basolite® F300 were all purchased from Sigma Aldrich. 1,3,5-benzentricarboxylic-d6 acid was obtained through the ISIS Neutron and Muon Source deuteration facility.

Fe-BTC was synthesised following a procedure adapted from ref. [21]. Iron (III) nitrate nonahydrate (2.5988 g, 6.43 mmol) and 1,3,5-benzenecarboxylic acid (1.1770 g, 5.60 mmol) were each dissolved in 20 mL of methanol. The two solutions were combined at room temperature and stirred vigorously. Within minutes the reaction mixture appeared to gelate, the viscosity rapidly increased until a non-flowing gel had formed. The gel was left to age at room temperature for 24 h. It was then washed with ethanol (3 × 20 mL) before drying at 60 °C overnight and ground to a fine, orange, powder. The powder was then purified in accordance to ref. [61] which reported to increase the surface area. Briefly, the powder was dispersed and heated for 3 h in each water (700 mL at 70 °C), ethanol (700 mL at 65 °C) then aqueous 38 mM ammonium fluoride solution (700 mL at 70 °C). The powder was recovered between each stage by centrifugation. The resultant product was dried overnight at 60 °C. A deuterated sample of Fe-BTC was synthesised by dissolving iron (III) nitrate nonahydrate (5.1976 g, 12.9 mmol) and 1,3,5-benzentricarboxylic-d6 acid (2.4216 g, 11.2 mmol) each in 40 mL of methanol-OD. An additional 200 mL of methanol-OD was required to dissolve the deuterated linker. The solutions were combined and left to stir at room temperature for 24 h. The gel was then washed and purified as above. Deuterium oxide was used in place of water when required.

MIL-100 (Fe), $Fe_3O(F)(H_2O)_2[(C_6H_3)(CO_2)_3]_2.nH_2O$, was synthesised following the procedure in ref. [25]. Briefly, 1,3,5-benzenetricarboxylic acid (1.676 g, 7.98 mmol) was dissolved in an aqueous 1 M solution of sodium hydroxide (23.72 g) and iron (II) chloride tetrahydrate (2.260 g, 11.4 mmol) separately dissolved in water (97.2 mL). The linker solution was then added dropwise to the

metal solution. A green suspension formed immediately, which slowly turned brown over 24 h whilst stirring at room temperature. The suspension was centrifuged, and the resulting powder washed thoroughly with ethanol (3 × 20 mL). The powder was then purified and dried as described above.

**Powder X-ray diffraction**. Data were collected at room temperature using a Bruker D8 diffractometer using Cu Kα1 ($\lambda = 1.5406$ Å) radiation and a LynxEye position-sensitive detector with Bragg-Brentano parafocusing geometry. Samples of finely ground powder were dispersed onto a low background silicon substrate and loaded onto the rotating stage of the diffractometer. Data were collected over the scattering angle range $2° < 2\theta < 45°$. Pawley refinements were carried out using TOPAS Academic (V6) software[62]. The unit cell parameters were refined against those previously reported for MIL-100 (Fe)[28]. A modified Thompson-Cox-Hasting pseudo-Voigt peak shape and simple axial divergence correction were employed. Refinements were carried out over the angular range $2° < 2\theta < 45°$.

**Fourier-transform infrared spectroscopy**. A Bruker Tensor 27 Infrared Spectrometer was used to collect Fourier-transform infrared spectroscopy data. Data were collected in transmission mode between 590 and 4000 cm$^{-1}$. For fast data collection, the spectrometer was fitted with an attenuated total reflectance cell. A background was collected and subtracted from all spectra.

**Elemental analysis**. Analysis was conducted using an Exeter Analytical CE-440 Elemental Analyser with gas mixtures analysed using thermal conductivity detectors. Samples (*ca.* 1.8 mg) were weighed into tin capsules and sealed before analysis.

**Helium pycnometry**. Sample densities were measured using a Micromeritics Accupyc 1340 helium pycnometer with a cylindrical 1 cm$^3$ insert. Typically, between 100 and 200 mg of sample was used, and the cycle involved 10 measurements. Prior to the measurement, samples were degassed at 150 °C under dynamic vacuum overnight.

**Thermogravimetric analysis**. Analysis was performed on a TA Instruments TGA-Q400, under an air atmosphere, using a heating rate of 10 °C min$^{-1}$, from room temperature up to 850 °C. For each measurement around 10 mg of evacuated sample was used.

**Nitrogen porosimetry**. Measurements were carried out at 77 K using a Micromeritics TriStar II instrument. Approximately 40 mg of each sample was degassed at 120 °C for 12 h prior to measurement. The specific surface area of the materials were calculated using the multipoint Brunauer–Emmett–Teller (BET) method applied to the isotherm adsorption branch, while taking into account the Rouquerol consistency criteria[63].

**Scanning electron microscopy**. Microscopy was carried out on a ZEISS Crossbeam 540 equipped with a Gemini 2 column. Finely ground samples were dispersed onto carbon tape on an aluminium specimen stub. A 1 kV accelerating voltage was used at a working distance of approximately 4.5 mm, the secondary electron signal was collected.

**High-resolution scanning transmission electron microscopy**. Aberration corrected high-resolution scanning transmission electron microscopy (HR-STEM) was carried out using a JEOL ARM300CF microscope equipped with a cold field emission electron source and JEOL aberration correctors in both the probe-forming and image-forming optics, located in the electron Physical Sciences Imaging Centre (ePSIC) at the Diamond Light Source, UK. Finely ground powder samples were dispersed by drop-casting from a suspension in methanol onto lacey carbon grids. The microscope was operated at 200 kV with a convergence semi-angle of approximately 14 mrad. Micrographs were recorded using the annular bright-field (ABF) STEM signals obtained using a combination of an ABF aperture and bright-field detector, with a camera length selected such that the outer edge of the direct beam disk was aligned with the outer edge of the ABF aperture. The inner angle of the ABF aperture was approximately 50% of the outer angle, giving a collection window of ~7–14 mrad. Annular dark-field (ADF) STEM images were acquired simultaneously with an inner collection angle >40 mrad.

**Small- and wide-angle X-ray scattering**. Combined SAXS and WAXS data were collected at the I22 beamline (Diamond Light Source, UK) using synchrotron radiation ($\lambda = 0.9998$ Å, 12.401 keV). The SAXS detector was positioned at a distance of 9.72623 m from the sample with a $Q$ range of 0.0014 to 0.178 Å$^{-1}$. The WAXS detector was positioned at a distance of 0.1679 m from the sample with a $Q$ range of 0.223 to 5.72 Å$^{-1}$. Samples were dispersed between two layers of transparent Scotch Tape and cut, using a circular die, to a diameter of 5 mm. Samples were measured at room temperature and the raw scattering data were reduced to one dimensional datasets using DAWN[64].

**X-ray total scattering**. Data were collected at the I15-1 beamline (Diamond Light Source, UK) using synchrotron radiation ($\lambda = 0.161669$ Å, 76.7 keV). A small amount of finely ground sample was loaded into a borosilicate capillary (inner diameter of 1.17 mm) to a height of 3.5 cm. Capillaries were sealed and mounted onto the instrument. Data were collected under ambient conditions for each sample, an empty capillary and the blank instrument over the region $\sim 0.4 < Q < \sim 26$ Å$^{-1}$. The raw total scattering data were corrected for background, multiple scattering, container scattering and Compton scattering along with absorption corrections, using GudrunX following standard procedures over the range $\sim 0.6 < Q < \sim 23$ Å$^{-1}$ [65]. The Fourier transform was then calculated to produce the pair distribution function, $G(r)$. We also make use of the $D(r)$ formulation of the pair distribution function to accentuate correlations at high-$r$[40].

**Neutron total scattering**. Data from Fe-BTC were measured on the neutron time-of-flight diffractometer GEM (ISIS Neutron and Muon Source, UK). A finely ground sample of deuterated and evacuated Fe-BTC was placed inside a cylindrical vanadium can of ~60 mm length and 8 mm diameter. Data were collected under ambient conditions for the sample, empty vanadium can, vanadium/niobium standard and empty instrument over the region $\sim 0.6 < Q < \sim 40$ Å$^{-1}$. The raw total scattering data were corrected using the GundrunN software, based on the Atlas routines, which also merges the data from different detector banks into a single normalised total scattering structure factor[65]. The Fourier transform of the merged total structure factor was calculated to obtain the pair distribution function.

We previously reported the following data on MIL-100 in ref. [34], and is reproduced here for clarity: Fourier-transform infrared spectroscopy, elemental analysis, helium pycnometry, thermogravimetric analysis, nitrogen adsorption porosimetry, scanning electron microscopy, high-resolution scanning transmission electron microscopy and X-ray total scattering data. All data on Fe-BTC and those not mentioned above for MIL-100 are unique to this study. All characterisations were carried out on the same batch of samples.

**Structural modelling**. Amorphous models were constructed using the Polymatic algorithm[17]. Firstly, a periodic simulation box, of length between 80 and 90 Å, was populated with linker units and either trimer units, tetrahedral assemblies or a combination of both to give initial models with 18,800 atoms. Bonds were then formed between reactive sites on opposite building blocks within a cut-off distance of 5 Å. Coupled with molecular dynamics steps, this stage was repeated until no further bonds could be created. After polymerisation, the unreacted sites were saturated with capping groups. The structure was then annealed using a 21-step molecular dynamics equilibration, to produce a physically sensible structure. Porosity of the final structures was evaluated using Zeo $++$[43]. Accessible and non-accessible geometric surface area was calculated using a nitrogen probe with a radius of 1.82 Å. Pore size distributions were calculated using a probe radius of 0.1 Å. Visual pore size distributions were calculated using a probe radius of 1.2 Å. For further details on model generation, validation and properties see, Supplementary Methods (Supplementary Figs. 22–27 and Supplementary Tables 6–10).

Partial pair distribution functions, $g_{ij}(r)$, were calculated from the reported MIL-100 structure and the amorphous models using the RMCProfile software package[28,66]. To obtain the total pair distribution functions, the Fourier transform of each $g_{ij}(r)$, yielding the partial structure factor, $A_{ij}(Q)$, was calculated. The $A_{ij}(Q)$ functions were then multiplied by their respective X-ray ($Q$-dependent) or neutron weighting coefficients and summed to obtain the total structure factor, $F(Q)$. The Fourier transform of $F(Q)$ was then calculated to obtain the total pair distribution function, $G(r)$.

## Data availability
The experimental data that support the findings of this study are available at https://doi.org/10.17863/CAM.63689. Representative configurations of the three model types can be found at https://github.com/Ibechis/FeBTC_models.

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

## Acknowledgements
A.F.S. acknowledges the EPSRC for a PhD studentship award under the industrial CASE scheme, along with Johnson Matthew PLC (JM11106). T.D.B. thanks the Royal Society for a University Research Fellowship (UF150021), the Leverhulme Trust for a Philip Leverhulme Prize, and the University of Canterbury Te Whare Wānanga o Waitaha, New Zealand, for a University of Cambridge Visiting Canterbury Fellowship. K.E.J. thanks the Royal Society for a University Research Fellowship and the European Research Council under FP7 (CoMMaD, ERC Grant No. 758370). S.M.C. acknowledges support from the Henslow Research Fellowship at Girton College, Cambridge. The authors would like to thank the JMTC analytical department and in particular Przemyslaw Magdziarz for CHN analysis. The ZEISS Crossbeam SEM is funded by the Henry Royce Institute Equipment Grant EP/P024947/1. We extend our gratitude to Diamond Light Source, Rutherford Appleton Laboratory, U.K., for access to beamline I15-1 (EE20038-1), I22 (SM24563-1) and ePSIC (EM20198-7). In particular we thank Dr. Mohsen Danaie for his assistance at ePSIC. We extend our gratitude to ISIS Neutron and Muon Source, Rutherford Appleton Laboratory, U.K., for access to the GEM instrument (RB1920007, Data https://doi.org/10.5286/ISIS.E.RB1920007). We would like to thank Dr Sarah Youngs for the synthesis of deuterated trimesic acid.

## Author contributions
A.F.S. synthesised and characterised the samples, and carried out the X-ray diffraction measurements; A.F.S., D.N.J. and S.M.C. performed high-resolution scanning transmission electron microscopy; A.F.S. and G.D. performed scanning electron microscopy; T.J. measured nitrogen adsorption isotherms and A.F.S. performed the analysis; A.F.S. and A.J.S. carried out combined small- and wide-angle X-ray scattering measurements; A.F.S. and P.A.C. carried out X-ray total scattering measurements; A.F.S. and D.A.K performed and interpreted X-ray pair distribution function analysis; A.F.S. synthesised the deuterated sample; A.F.S., D.A.K. and T.D.B. collected neutron total scattering data; D.A.K. carried out neutron pair distribution function analysis; M.A. developed the force field used in the simulations; I.B. and K.E.J. generated the amorphous structural models and performed computational porosity measurements; A.F.S. wrote the paper with input from all authors.

## Competing interests
T.J. works for Johnson Matthey PLC, a company with interest in commercialisation of MOF materials. The remaining authors declare no competing interests.
