## [Peer Review File · Nature Communications]

REVIEWER COMMENTS

Reviewer #1 (Remarks to the Author):

The authors report atomic-scale structure of a metal-organic framework, Fe-BTC by using advanced electron microscopy, synchrotron radiation and polymerisation-based algorithm calculation. Although Fe-BTC is one of the very few commercially available MOFs and it has morphologically diverse family of materials, atomic-scale structure of Fe-BTC still remains unknown. This paper potentially presents a convincing solution of the atomic-scale structure of Fe-BTC material. This paper might be acceptable for publication in Nature Communications after providing the answers to the following comments.

(a) About the structural modelling: Based on the similarity in PDF peak analysis and experimental porosity analysis, the authors claim that MIX (MRO) model best reflects the atomic-scale structure for Fe-BTC (Basolite F300). I think that this claim excludes the possibility of the mixture of several phase (in this case SRO, MIX, and MRO). How did the authors confirm that the both synthesized Fe-BTC and obtained Basolite F300 are a single phase? The XRD analysis, SEM micrographs, and elemental analysis data presented here are not enough for the determination of single phase of the present compounds. Especially, TEM images for Fe-BTC seem to be multi-phase rather than single phase. I do not agree that both Fe-BTC and Basolite F300 are in a single phase.

(b) I strongly recommend the authors to discuss this point in detail. Also, the surface area calculated for the MIX model (Figure 3a) is not consistent with the experimental result (non-porous character) from nitrogen adsorption isotherm. How does the authors explain about this?

(c) In Figures S1 and S2, the authors show Pawley refinement results for MIL-100 (Fe) and Fe-BTC powder. However, the authors did not show fitting results such as lattice constants, reliability factors, and S values for both samples. Also, in the elemental analysis data (Table S3), the authors do not show any empirical formula for MIL-100 (Fe) and Fe-BTC. Please provide these data.

(d) In this paper, the authors do not show any data from fitting analysis on the PDF patterns. I strongly recommend the authors to add the more quantitative analysis for PDF analysis to confirm the validity. For example, multi-phase analysis using SRO, MIX, and MRO models and fitting analysis for total pair distribution function, $G(r)$.

(e) I also request the authors to show the crystal structure file from PDF analysis and theoretical modelling for Fe-BTC as .cif, .pdb, and .xyz files including the lattice parameters, atomic coordinates, and bond lengths. This kinds of information should be added in Supplementary Information.

Reviewer #2 (Remarks to the Author):

This work challenges the characterization of amorphous Fe-BTC compounds. The study involves many expensive techniques and modelling including: Synchrotron X-ray, HR-STEM, and a polymerisation-based algorithm, and succeeded in illustrating the proposed structures. As they have studied the structure, they found the sample is the mixture of crystal and amorphous domains. This finding made the challenge harder, and significantly influenced the discussion. As they described, the amorphous Fe-BTC is usually made via sol-gel process. Reviewer could expect the potential concern of sample homogeneity, as BET surface areas varies depend on the synthetic protocols. All the characterization should be conducted with same batch samples.

The authors insisted the polymerization-based algorithm could be applied to other amorphous MOFs. So far, only ZIF-4 and present Fe-BTC have been the limited examples both of which have high structural symmetry. Could they provide some comparative interpretation for cases of ZIF-4 and Fe-BTC contributed by the algorithm? Reviewer is sure ZIF-4 does not have mixture of crystal and amorphous domains and much simple connectivity toward metal ions.

Mixing of two phases (crystal and amorphous) affects the discussion of the ratio and connectivity of trimer and tetrahedra in the structure in page 6. As they pointed out, the discussion of the ratio in MIL-100 and Fe-BTC is confusing for the mixture samples. One of main curiosities is the volumetric ratio of crystalline and amorphous parts as well as interface structure of these two phases. Could they add discussion on this points from the careful views of HR-TEM?

Amorphous Fe-BTC does not show any uptake of N₂ at 77 K. This is because of the molecular clog at the restricted pore interior. Did they measure Ar sorption at 87 K? Figure 3b shows the 3D pore distribution with pore radius. The structures look like having bottle and neck open channels and it is related to the gas (and ion) diffusion. Could they visualize the connected pore structures from the model analysis?

Although they often mention that Fe-BTC proposed amorphous structures would work for heterogeneous catalysis or ion transporter media, they did not study these properties at all. Reviewer recommends reducing these statements. Besides, researchers know the catalytic sites of MIL-100 is attributed from trimer unsaturated metal center and it is attractive how the active sites are preserved in the disordered structures and probability of redox activity in Fe ions.

Reviewer #3 (Remarks to the Author):

The authors of this article characterize the very-difficult-to-characterize amorphous Fe-BTC structure using several first-of-a-kind techniques and show their results. I found the article to be very well written, clearly presented, and consequently only have a few minor critiques which I list below roughly in the order they appear in the paper:

- Given the frequent mention of the commercial Basolite material, and that one of the authors (Timothy Johnson) disclosed a financial conflict of interest, I think it would be helpful and appropriate for Timothy to disclose the specific company rather than just "a MOF company." Conflicts of interest are totally fine in this kind of work but being mysterious and cagey about it only raises suspicions.
- On line 44 the first sentence refers to probes "commonly used in the characterization of amorphous MOFs." This sentence is rather at odds with the earlier paragraph that talks about how rare amorphous MOF structure characterization of any kind is (specifically the word "commonly"). Perhaps the authors can clarify this point, especially as it pertains to the novelty of the whole paper.
- It may just be a matter of personal preference, but several Figures in the paper force the reader to look back and forth between the caption and the plots in a way that seems unnecessary except in the pursuit of some ideal of aesthetic minimalism. For example, Fig1e, Fig 4b, and Fig 4d, all show two curves with different colors, but instead of labelling them on the plot itself one has to consult the caption to understand which is which. At the risk of violating someone's style guide, I think it would be easier to digest the contents of the paper if an additional label or two were added to these plots.
--- Fig 3b would also benefit from a better indication of scale. How large are these simulation boxes? I think I found 80/85 angstrom box length number in the Supporting Information, but that was bit hard to find relative to how easy it would have been if it were mentioned on that Figure.
- There are a few references to Basolite F300's "diverse catalytic ability" but without any further elaboration. I recognize that the point of the paper is not catalysis, but since this is mentioned more than once as a motivation for this work, it would be nice if an additional sentence or two could be added to indicate what particular catalytic reactions this material is used for (and more to the point, how a better understanding of the amorphous structure can help further improve those applications).
- One person in the field who I always feel has never received sufficient credit for her work is Caroline Mellot-Draznieks. The method of the authors in the present work, which combines Polymatic but uses MOF-based building units, is very closely related to the methodology that Dr. Mellot-Draznieks developed in generating hypothetical MOFs ~15 years ago (it was called the "AASBU" method). (Except that here the systems are amorphous and hers were crystalline!). How closely related these methods are is ultimately subjective and I leave it up to the authors to decide

whether to mention this in their paper or not.

- As you may be able to tell from my comments above, I have struggled tremendously to find reasons to be critical of this excellent work. I look forward to seeing it published and am excited to see what the authors come up with next.

REVIEWER COMMENTS

Reviewer #1 (Remarks to the Author):

The authors report atomic-scale structure of a metal-organic framework, Fe-BTC by using advanced electron microscopy, synchrotron radiation and polymerisation-based algorithm calculation. Although Fe-BTC is one of the very few commercially available MOFs and it has morphologically diverse family of materials, atomic-scale structure of Fe-BTC still remains unknown. This paper potentially presents a convincing solution of the atomic-scale structure of Fe-BTC material. This paper might be acceptable for publication in Nature Communications after providing the answers to the following comments.

We would like to thank the reviewer for their time and are pleased they consider the manuscript suitable for publication in Nature Communications following minor revisions.

(a) About the structural modelling: Based on the similarity in PDF peak analysis and experimental porosity analysis, the authors claim that MIX (MRO) model best reflects the atomic-scale structure for Fe-BTC (Basolite F300). I think that this claim excludes the possibility of the mixture of several phase (in this case SRO, MIX, and MRO). How did the authors confirm that the both synthesized Fe-BTC and obtained Basolite F300 are a single phase? The XRD analysis, SEM micrographs, and elemental analysis data presented here are not enough for the determination of single phase of the present compounds. Especially, TEM images for Fe-BTC seem to be multi-phase rather than single phase. I do not agree that both Fe-BTC and Basolite F300 are in a single phase.

The reviewer is correct on all accounts here. Our modelling approach sought to capture the average structure of the bulk amorphous phase of the materials in both cases. The main challenge associated with modelling amorphous materials is that an exact atomic configuration cannot be deduced. Instead, we seek to obtain large atomic ensembles that are representative of the bulk structure. It is often noted that such models may not be large enough to fully encompass the true structural diversity associated with the material. In this sense, it may well be the case that Fe-BTC and Basolite F300 have structures intermediate to the three models presented here. In the text we note that our analysis describes the average structure and that there is likely to be a wide structural variety in the material (lines 285–6). We have updated this to more explicitly describe some of the structural variety anticipated.

Of course, given the presence of both crystalline and amorphous domains in Fe-BTC, these values only represent an average, and it is likely that there will be a range of microscopic heterogeneity within the material.

The multi-phase modelling the reviewer is suggesting is an excellent idea and is something we touch on in the discussion section of the manuscript (lines 412–419). Models of this nature are not impossible to produce, however they come with several computational challenges – such as structural relaxation at the interfacial boundary – that are beyond the scope of this work. We do agree that the potential for a phase mixture between the SRO, MIX and MRO models is a particularly interesting avenue for future investigation and this has therefore been acknowledged in the discussion section of the manuscript.

We note that while the models here represent the bulk amorphous structure, it is possible that there may also be phase mixing amongst the three model types, even on a microscopic scale. There is, of course, an entire continuum of configurations between the extremes of our SRO and MRO models.

On the reviewers second point, regarding the phase purity of the materials, they are entirely correct. A crucial implication of our study is that X-ray diffraction and other standard characterisation techniques are often insufficient to truly capture the complexity of these materials. Our combined use of SEM and HR-STEM (as a local crystallinity probe) is pivotal here in identifying the nanocomposite structure of Fe-BTC. We believe this is what differentiates our work from many other studies on Fe-BTC, to the best of our knowledge this is the first description of heterogenous, nanocomposite phase behaviour in Fe-BTC.

Hence, we agree that Fe-BTC as synthesised here is not a single phase, but possesses a nanocomposite structure, something that may have been overlooked in previous studies. As we discuss later in response to reviewer 2, from our SEM and HR-STEM analysis we deduce that the crystalline component of the nanocomposite structure represents a very small fraction of the overall structure and hence our efforts to capture the bulk amorphous structure of Fe-BTC are valid. Furthermore, our analysis of the PDF revealed that the local structure of nanocomposite Fe-BTC can be described by the trimer and tetrahedral building units. In other words, the crystalline and amorphous phase have the same local structure and the main differentiating feature is the presence of long-range order in the crystalline phase. In order to clarify the above, we have included the following text highlighting our intention to model the bulk, amorphous structure of Fe-BTC.

Polymatic, a polymerisation-based algorithm, was used to generate atomistic models of the amorphous phase that represents the bulk of Fe-BTC.

(b) I strongly recommend the authors to discuss this point in detail. Also, the surface area calculated for the MIX model (Figure 3a) is not consistent with the experimental result (non-porous character) from nitrogen adsorption isotherm. How does the authors explain about this?

The surface areas reported in 3a are calculated from the models using a nitrogen-sized probe. The figure shows the contribution from accessible and non-accessible surface area towards the nitrogen probe. For the MIX model, we find it contains only non-accessible surface area that would not be detected using a standard nitrogen adsorption experiment. Hence, the MIX model is consistent with the non-porous character observed experimentally for Fe-BTC. It is only the MRO model that contains accessible surface area which would be experimentally measured using a nitrogen adsorption experiment. This has now been clarified in the text.

Both the SRO and MIX models contained only non-accessible surface areas of 243(35) and 392(36) $\text{m}^2 \text{g}^{-1}$ respectively, which would not be detected using a nitrogen adsorption experiment.

(c) In Figures S1 and S2, the authors show Pawley refinement results for MIL-100 (Fe) and Fe-BTC powder. However, the authors did not show fitting results such as lattice constants, reliability factors, and S values for both samples. Also, in the elemental analysis data (Table S3), the authors do not show any empirical formula for MIL-100 (Fe) and Fe-BTC. Please provide these data.

We are grateful to the reviewer for pointing out these omissions. We have now included the following table for the Fe-BTC refinement details in the Supplementary information.

Table S1 Crystallographic data from Pawley refinement of Fe-BTC. It is important to note that little physical significance is given to this refinement as the peaks are much broader than the typical separation of MIL-100's Bragg peak positions and it is simply illustrating that such diffraction data can lead to successful convergence of a refinement when considering such small domain sizes.

$R_{wp} = 7.50$	Experimental
$a = b = c$	73.234(4)
$\alpha = \beta = \gamma$	90

The theoretical elemental analysis results have been calculated from the empirical formula for MIL-100 and included in the Supplementary Information table. The following comments have been added to the discussion of the elemental analysis and TGA data in the manuscript.

Table S2 Elemental analysis of MIL-100 and Fe-BTC and theoretical values calculated from the empirical formula for MIL-100, $\text{Fe}_3\text{O}(\text{F})(\text{H}_2\text{O})_2[(\text{C}_6\text{H}_3)(\text{CO}_2)_3]_2$.

	MIL-100	Fe-BTC	MIL-100 calculated [1]
C wt.%	31.2	33.6	33.1
H wt.%	2.4	2.41	1.5
N wt.%	0.24	0.57	0

Elemental (CHN) analysis revealed that Fe-BTC and MIL-100 have very similar chemical compositions, containing 33.6 and 31.2 wt.% of carbon respectively which is in line with the theoretical 33.1 wt.% calculated from the empirical formula for MIL-100 [Table S4].

The iron content in MIL-100 and Fe-BTC was estimated as 26.1 and 21.8 wt.% respectively, assuming hematite to be the only solid product remaining after heating to 850 °C, which is close to the expected 25.7 wt.% calculated from the empirical formula for MIL-100.

(d) In this paper, the authors do not show any data from fitting analysis on the PDF patterns. I strongly recommend the authors to add the more quantitative analysis for PDF analysis to confirm the validity. For example, multi-phase analysis using SRO, MIX, and MRO models and fitting analysis for total pair distribution function, $G(r)$.

As previously mentioned in (a), the multi-phase modelling is a great suggestion. The beauty of our Polymatic routine is that fitting analysis of this kind is not required. Unlike reverse Monte Carlo (RMC) approaches, where fitting of the model to the data is necessary, Polymatic allows for the generation of amorphous structures without any experimental input. RMC methods cannot be used in the same predictive way as Polymatic may enable, as described in the discussion section.

The amorphous models generated here could, in theory, be used as starting models for RMC refinement of our experimental data. This in itself represents a significant challenge that we are starting to pursue. However, we believe that the inclusion of such techniques here would detract from the novelty of our work – the ability to generate amorphous networks in the absence of experimental data and the potential for this approach to be used in a predictive nature to discover new functional, amorphous, hybrid materials. To highlight our intentions to explore this interesting realm of computational modelling further, we have amended the discussion as follows.

On a structural level, our immediate goal would be to characterise the crystalline domains to obtain a model of the atomic structure of these regions through electron diffraction measurements for example. Subsequent combination of the crystalline model with that of our amorphous phase presented here would provide a model for the full nanocomposite structure of Fe-BTC. This model could then serve as a starting configuration for a large-box refinement against the experimental data. Both stages represent significant challenges; structure solution within composite materials is very difficult and has yet to be reported using electron diffraction measurements, which are typically limited to single crystal samples.⁴⁵ Furthermore, while two-phase modelling of nanocomposite structures is possible, the RMC methods required are complex.³⁹ In particular, structural relaxation at the interfacial boundary between phases and energetic relaxation of the final configuration are both very challenging, yet exciting, avenues for future exploration. We note that while the models here represent the bulk amorphous structure, it is possible that there may also be phase mixing amongst the three model types, even on a microscopic scale. There is of course an entire continuum of configurations between the extremes of our SRO and MRO models.

Reference 45 was also updated to the following, more appropriate, review on electron diffraction techniques for structure solution.

45. Yun, Y., Zou, X., Hovmöller, S. & Wan, W. Three-dimensional electron diffraction as a complementary technique to powder X-ray diffraction for phase identification and structure solution of powders. *IUCr* **2**, 267–282 (2015).

(e) I also request the authors to show the crystal structure file from PDF analysis and theoretical modelling for Fe-BTC as cif, pdb, and xyz files including the lattice parameters, atomic coordinates, and bond lengths. This kind of information should be added in Supplementary Information.

We thank the reviewer for this suggestion, the models produced by Polymatic have been uploaded to the GitHub data repository and the link has been included in the Additional Information section of the manuscript. However, as outlined above we have not performed modelling of the PDF data and have therefore not included refined structures.

Reviewer #2 (Remarks to the Author):

This work challenges the characterization of amorphous Fe-BTC compounds. The study involves many expensive techniques and modelling including: Synchrotron X-ray, HR-STEM, and a polymerisation-based algorithm, and succeeded in illustrating the proposed structures. As they have studied the structure, they found the sample is the mixture of crystal and amorphous domains. This finding made the challenge harder, and significantly influenced the discussion. As they described, the amorphous Fe-BTC is usually made via sol-gel process. Reviewer could expect the potential concern of sample homogeneity, as BET surface areas varies depend on the synthetic protocols. All the characterization should be conducted with same batch samples.

We thank the reviewer for their comments, in particular for their appreciation of the challenging nature of the work presented. This is an excellent point regarding homogeneity and is something that might often be ignored in the study of these complex materials. Minor synthetic variations in Fe-BTC synthesis have produced a large diversity of materials. As discussed in the introduction to the manuscript, these variations can lead to the formation of aerogel, xerogel, powdered and even crystalline materials. We anticipated that sample homogeneity may likely be a concern and hence our Fe-BTC and MIL-100 syntheses were carried out on a gram-scale to afford sufficient product to enable

all characterisation to be carried out on the same batch of material. This has been made clear in the methods section.

All characterisations were carried out on the same batch of samples.

The authors insisted the polymerization-based algorithm could be applied to other amorphous MOFs. So far, only ZIF-4 and present Fe-BTC have been the limited examples both of which have high structural symmetry. Could they provide some comparative interpretation for cases of ZIF-4 and Fe-BTC contributed by the algorithm? Reviewer is sure ZIF-4 does not have mixture of crystal and amorphous domains and much simple connectivity toward metal ions.

The reviewer is correct, both crystalline ZIF-4 and MIL-100 have high structural symmetry. The models of ZIF-4 and Fe-BTC generated by Polymatic, however, are amorphous and contain no long-range structural symmetry. Importantly, our approach is entirely independent of the crystal symmetry of the crystalline framework. The Polymatic algorithm requires only the structural building units of the MOF, which are randomly placed inside the simulation box, and is not influenced by the original crystal symmetry. Hence, it is possible to generate amorphous models from crystalline MOFs that have low structural symmetry as this plays no role in the Polymatic algorithm. This has now been made clear in the Introduction and Structural Modelling sections. We also agree that crystalline and amorphous ZIF-4 are both single phase materials and are comprised of much simpler structural building units. Fe-BTC, on the other hand, with its complex phase behaviour and hierarchical local structure represents a greater challenge, which ultimately led us to study this fascinating material in great detail.

The structural building blocks of ZIF-4 were polymerised together to generate an amorphous model through a process that was entirely independent of the crystal symmetry of crystalline ZIF-4.

Polymatic allows for the polymerisation of monomer units – metal nodes (trimer units and tetrahedral assemblies) and 1,3,5 benzenetricarboxylate anions in this case – into amorphous models.

In terms of a comparison between the two, one thing that springs to mind is the degree of polymerisation. For example, in ZIF-4 a 98.6% degree of polymerisation was achieved in comparison to 92% in Fe-BTC. This reduction in polymerisation was, as the reviewer stated, due to the complexity of the structural building units in Fe-BTC. We have made note of this comparison within the Structural Modelling section.

A total of approximately 92% of reactions were completed in these simulations in comparison to 98.6% obtained for α ZIF-4, this reduction was due to the increased complexity and size of the structural building units in Fe-BTC compared to ZIF-4.¹⁸

In addition, we have also made clear that the previous model of amorphous ZIF-4, produced using Polymatic, was not compared to experimental pair distribution function data nor the amorphous ZIF-4 model obtained through RMC refinement. Our work here represents the first example of a Polymatic-derived MOF model being scrutinised on this structural level.

Despite the advantages of polymerisation-based modelling, to the best of our knowledge it has currently only been used to produce a model for the α ZIF-4 structure, though this was not experimentally verified using pair distribution function data nor compared to the RMC-derived model for α ZIF-4.

Mixing of two phases (crystal and amorphous) affects the discussion of the ratio and connectivity of trimer and tetrahedra in the structure in page 6. As they pointed out, the discussion of the ratio in

MIL-100 and Fe-BTC is confusing for the mixture samples. One of main curiosities is the volumetric ratio of crystalline and amorphous parts as well as interface structure of these two phases. Could they add discussion on these points from the careful views of HR-TEM?

We agree that the interaction between the phases is particularly interesting, especially considering the increased likelihood for structural defects to occur at the interfacial boundary. Precise calculation of the volumetric ratio was not possible due to the irregularity of the particle shapes and sizes and the limited resolution of the images. HR-STEM is further complicated by the necessity for planes in a lattice to be near-perpendicular to the beam and distinguishable by the resolution of the image. As a result, the images of fringes are acquired at a magnification that is not directly compatible with large fields of view over significant fractions of the material. However, close inspection of our SEM images showed the amorphous larger fragments were the dominant phase, accounting for the majority of the sample, which we have now made clear in the text.

Visual inspection of the images indicated the majority of the sample comprised of the larger fragments, suggesting this is the dominant phase.

We have expanded the discussion of the HR-STEM data and have now included reference to the interfacial region.

These periodic features were typically observed over a much smaller field of view than in MIL-100, within the region of 10 to 50 nm wide, and likely correspond to the nanoparticles observed in the scanning electron microscopy.

At the interfacial region between phases, the contrast was often mottled, and clear boundaries were hard to identify suggesting intimate phase interactions. Collectively, the electron microscopy data suggests a nanocomposite structure of Fe-BTC containing two distinct phases: an amorphous matrix representing the vast majority of the material, and crystalline nanoparticles of a phase other than MIL-100.

Amorphous Fe-BTC does not show any uptake of N_2 at 77 K. This is because of the molecular clog at the restricted pore interior. Did they measure Ar sorption at 87 K?

This a good suggestion due to the smaller molecular dimensions of Ar compared to N_2 . We did collect Ar sorption data for both MIL-100 and Fe-BTC at 273 K up to 100 kPa (see Reviewer Figure 1 below). However, we found that both materials exhibited near-identical behaviour, showing little favourable interaction with the frameworks. These isotherms did not significantly improve our understanding of the structure of Fe-BTC and hence they were not included in the manuscript.

We do agree with the reviewer that additional investigation of the porous interior is an interesting idea for further research, unfortunately we are currently unable to obtain further sorption measurements at the moment. However, it is an avenue that we will be actively pursuing when we are able to collect new experimental data.

Reviewer Figure 1 Adsorption (closed symbol) and desorption (open symbol) isotherms of Ar in MIL-100 (blue) and Fe-BTC (orange).

Figure 3b shows the 3D pore distribution with pore radius. The structures look like having bottle and neck open channels and it is related to the gas (and ion) diffusion. Could they visualize the connected pore structures from the model analysis?

The structure of the porous interior is indeed a fascinating feature of our models, which we agree isn't fully highlighted by Figure 3b alone. We have therefore added the following text to the manuscript and figure to the Supplementary Information.

Visualisation of these nitrogen-probed surface areas revealed pore structures that are twisted and irregular in shape [Fig. S14].

Figure S14 The accessible (blue) and non-accessible (red) surface area to a nitrogen probe (3.64 Å diameter) for representative examples of the (a) SRO, (b) MIX and (c) MRO models.

Although they often mention that Fe-BTC proposed amorphous structures would work for heterogeneous catalysis or ion transporter media, they did not study these properties at all. Reviewer recommends reducing these statements. Besides, researchers know the catalytic sites of MIL-100 is attributed from trimer unsaturated metal centre and it is attractive how the active sites are preserved in the disordered structures and probability of redox activity in Fe ions.

We agree the presence of active sites in the disordered structures is very appealing. The statements regarding catalysis and ion transport have been less heavily emphasised. Taking reviewer 3's comments into consideration as well, we have expanded the initial discussion regarding the catalytic ability by giving specific examples and removed the subsequent mention on line 385.

For example, as a Lewis acid catalyst in Claisen-Schmidt reactions, ring opening of epoxides and selective hydrogenations, and as a catalyst in the oxidation of thiols and amines, achieving conversion rates and selectivities of 99% in some cases.

Reviewer #3 (Remarks to the Author):

The authors of this article characterize the very-difficult-to-characterize amorphous Fe-BTC structure using several first-of-a-kind techniques and show their results. I found the article to be very well written, clearly presented, and consequently only have a few minor critiques which I list below roughly in the order they appear in the paper:

We thank the reviewer for their time considering the manuscript and are very pleased they consider it suitable for publication in Nature Communications.

- Given the frequent mention of the commercial Basolite material, and that one of the authors (Timothy Johnson) disclosed a financial conflict of interest, I think it would be helpful and appropriate for Timothy to disclose the specific company rather than just "a MOF company." Conflicts of interest are totally fine in this kind of work but being mysterious and cagey about it only raises suspicions.

In the interest of transparency, the conflict of interest has been updated to the following.

T.J. works for Johnson Matthey PLC, a company with interest in commercialisation of MOF materials.

- On line 44 the first sentence refers to probes "commonly used in the characterization of amorphous MOFs." This sentence is rather at odds with the earlier paragraph that talks about how rare amorphous MOF structure characterization of any kind is (specifically the word "commonly"). Perhaps the authors can clarify this point, especially as it pertains to the novelty of the whole paper.

We agree with the reviewer that line 44 is slightly confusing and it has been changed to the following.

Local structure probes are often the only means for characterising the structure of amorphous MOFs.

- It may just be a matter of personal preference, but several Figures in the paper force the reader to look back and forth between the caption and the plots in a way that seems unnecessary except in the pursuit of some ideal of aesthetic minimalism. For example, Fig 1e, Fig 4b, and Fig 4d, all show two curves with different colors, but instead of labelling them on the plot itself one has to consult the caption to understand which is which. At the risk of violating someone's style guide, I think it would be easier to digest the contents of the paper if an additional label or two were added to these plots.

We appreciate the reviewer acknowledging the aesthetic nature of the manuscript. In the interest of clarity, additional labels have been added to Figures 1e, 4b and 4d.

- Fig 3b would also benefit from a better indication of scale. How large are these simulation boxes? I think I found 80/85 angstrom box length number in the Supporting Information, but that was bit hard to find relative to how easy it would have been if it were mentioned on that Figure.

We thank the reviewer for this good suggestion and have amended the caption to include the size of the simulation box, which is 70 Å. For completeness, we note that the 80–90 Å box size the reviewer quoted is the initial simulation box size, prior to polymerisation and annealing.

(b) Visual representation of the pore size distribution for the SRO (left), MIX (centre) and MRO (right) systems from a representative model of approximately 70 Å in length.

- There are a few references to Basolite F300's "diverse catalytic ability" but without any further elaboration. I recognize that the point of the paper is not catalysis, but since this is mentioned more than once as a motivation for this work, it would be nice if an additional sentence or two could be added to indicate what particular catalytic reactions this material is used for (and more to the point, how a better understanding of the amorphous structure can help further improve those applications).

We agree that our reference to Basolite F300's catalytic ability is rather vague and so have expanded our description in the introduction, noting several key examples, and discussion sections.

For example, as a Lewis acid catalyst in Claisen-Schmidt reactions, ring opening of epoxides and selective hydrogenations, and as a catalyst in the oxidation of thiols and amines, achieving conversion rates and selectivities of 99% in some cases.

On a more applied footing, the geometric analysis undertaken here on Fe-BTC suggests its suitability for ion transport, while the atomistic structure will be useful in understanding and improving the catalytic ability of Fe-BTC by helping to identify the presence of active sites for example.

- One person in the field who I always feel has never received sufficient credit for her work is Caroline Mellot-Draznieks. The method of the authors in the present work, which combines Polymatic but uses MOF-based building units, is very closely related to the methodology that Dr. Mellot-Draznieks developed in generating hypothetical MOFs ~15 years ago (it was called the "AASBU" method). (Except that here the systems are amorphous and hers were crystalline!). How closely related these methods are is ultimately subjective and I leave it up to the authors to decide whether to mention this in their paper or not.

The work of Dr. Mellot-Draznieks is very notable and we regret not having given her due credit in this work, indeed we have previously collaborated with her and respect her work enormously. We have now cited her following work on the *de novo* prediction of crystalline inorganic structures in the introduction.

Steps within the algorithm generated chemical bonds between reactive sites of the metal nodes and organic linkers, not too dissimilar to the automated assembly of secondary building unit method used in the *de novo* prediction of crystalline inorganic structures over 20 years ago.¹⁹

19. Mellot Draznieks, C., Newsam, J. M., Gorman, A. M., Freeman, C. M. & Férey, G. De novo prediction of inorganic structures developed through automated assembly of secondary building units (AASBU method). *Angew. Chemie - Int. Ed.* **39**, 2270–2275 (2000).

- As you may be able to tell from my comments above, I have struggled tremendously to find reasons to be critical of this excellent work. I look forward to seeing it published and am excited to see what the authors come up with next.

We are very grateful for the reviewer's kind comments and are happy they find the work of great interest.

REVIEWERS' COMMENTS

Reviewer #1 (Remarks to the Author):

The topic of this paper is atomic-scale structure of a metal organic framework, Fe-BTC by using advanced electron microscopy, synchrotron radiation and polymerisation-based algorithm calculation. I have just read the revised manuscript and I think the authors made an effort to respond to all the reviewers' comments. The requests made by the reviewers were addressed appropriately. Therefore, I think this paper is publishable in Nature Commun.

Reviewer #2 (Remarks to the Author):

The authors made actions to one-by-one points by the reviewers and added some more data of characterization and data (PDF detail, electron microscopy, Ar gas sorption, elemental analysis etc) to support their discussion.

Now it is publishable for Nature Comm. The authors should take care again of significant digits such in polymerization of monomer units, X-ray crystallography parameters.

The next step of this work will be about the elucidation of redox activity of Fe sites, porosity, and open metal sites (as discussed) related to the degree of amorphization. I look for expanding the modelling method to other MOFs, especially low-symmetric structures since the methodology does not depend on the structural symmetry which is a benefit compared with conventional DFT or MD.

REVIEWER COMMENTS

Reviewer #1 (Remarks to the Author):

The topic of this paper is atomic-scale structure of a metal organic framework, Fe-BTC by using advanced electron microscopy, synchrotron radiation and polymerisation-based algorithm calculation. I have just read the revised manuscript and I think the authors made an effort to respond to all the reviewers' comments. The requests made by the reviewers were addressed appropriately. Therefore, I think this paper is publishable in Nature Communications.

We would like to thank the reviewer for their time in reading the manuscript and are very pleased they believe it is suitable for publication in Nature Communications.

Reviewer #2 (Remarks to the Author):

The authors made actions to one-by-one points by the reviewers and added some more data of characterization and data (PDF detail, electron microscopy, Ar gas sorption, elemental analysis etc) to support their discussion. Now it is publishable for Nature Communications.

We are glad the reviewer is satisfied with our revised manuscript and believes it is ready for publication in Nature Communications.

The authors should take care again of significant digits such in polymerization of monomer units, X-ray crystallography parameters. The next step of this work will be about the elucidation of redox activity of Fe sites, porosity, and open metal sites (as discussed) related to the degree of amorphization. I look for expanding the modelling method to other MOFs, especially low-symmetric structures since the methodology does not depend on the structural symmetry which is a benefit compared with conventional DFT or MD.

We thank the reviewer for their astute observations and have corrected the number of significant digits as follows:

A total of approximately 91.9% of reactions were completed in these simulations in comparison to 98.6% obtained for α ZIF-4, this reduction was due to the increased complexity and size of the structural building units in Fe-BTC compared to ZIF-4

The value 91.9% is the average polymerisation value of the three model types (92.1, 92.2 and 91.5%).

Table S2 Crystallographic data from Pawley refinement of Fe-BTC. It is important to note that little physical significance is given to this refinement as the peaks are much broader than the typical separation of MIL-100's Bragg peak positions and it is simply illustrating that such diffraction data can lead to successful convergence of a refinement when considering such small domain sizes.

$R_{wp} = 7.50$	Experimental
$a = b = c$	73.24(4)
$\alpha = \beta = \gamma$	90

The number of significant digits has been changed so that it matches the crystallographic data for MIL-100.